# SDE-YOLO: A Novel Method for Blood Cell Detection

**DOI:** 10.3390/biomimetics8050404

**Published:** 2023-09-01

**Authors:** Yonglin Wu, Dongxu Gao, Yinfeng Fang, Xue Xu, Hongwei Gao, Zhaojie Ju

**Affiliations:** 1School of Automation and Electrical Engineering, Shenyang Ligong University, Shenyang 110158, China; wuyonglin2000@sohu.com (Y.W.); ghw1978@sohu.com (H.G.); 2School of Computing, University of Portsmouth, Portsmouth PO13HE, UK; 3School of Telecommunication Engineering, Hangzhou Dianzi University, Hangzhou 311305, China; yinfeng.fang@hdu.edu.cn; 4China Tobacco Zhejiang Indusirial Co., Ltd., Hangzhou 311500, China; xuxuehdck@sina.com

**Keywords:** blood cell testing, Swin Transformer, PAN, depth-separable convolution, EIOU

## Abstract

This paper proposes an improved target detection algorithm, SDE-YOLO, based on the YOLOv5s framework, to address the low detection accuracy, misdetection, and leakage in blood cell detection caused by existing single-stage and two-stage detection algorithms. Initially, the Swin Transformer is integrated into the back-end of the backbone to extract the features in a better way. Then, the 32 × 32 network layer in the path-aggregation network (PANet) is removed to decrease the number of parameters in the network while increasing its accuracy in detecting small targets. Moreover, PANet substitutes traditional convolution with depth-separable convolution to accurately recognize small targets while maintaining a fast speed. Finally, replacing the complete intersection over union (CIOU) loss function with the Euclidean intersection over union (EIOU) loss function can help address the imbalance of positive and negative samples and speed up the convergence rate. The SDE-YOLO algorithm achieves a mAP of 99.5%, 95.3%, and 93.3% on the BCCD blood cell dataset for white blood cells, red blood cells, and platelets, respectively, which is an improvement over other single-stage and two-stage algorithms such as SSD, YOLOv4, and YOLOv5s. The experiment yields excellent results, and the algorithm detects blood cells very well. The SDE-YOLO algorithm also has advantages in accuracy and real-time blood cell detection performance compared to the YOLOv7 and YOLOv8 technologies.

## 1. Introduction

Blood cell testing is a technique that analyses blood’s cellular components using specialist instruments. In medicine, the cells circulating in the blood are divided into three categories: white blood cells, red blood cells, and platelets [1]. When the results for the cell counts of these three types are abnormally high or low, the body may already suffer from certain diseases. The accurate and efficient identification of these three types of cells can provide a patient’s overall health status [2].

Previously, blood cell testing mainly used traditional image-processing techniques, most classically, manual photography and staining, followed by manual observation and classification with a microscope [3], which required the testers to have an extensive fundamental knowledge of cell morphology and to be skilled in morphological observation and discrimination through continuous study and repeated practice.

Due to the diversity of microscopic image data, traditional image-processing algorithms [4,5] can no longer meet the complex problems researchers encounter in cell detection and counting because of their low detection efficiency and low accuracy. Machine-learning algorithms for blood cell detection are later proposed [6,7,8,9,10,11]. Machine learning amalgamates fields like computer science, statistics, probability theory, etc. A mixture of computers and biological neural network structures enables computers to learn from vast data to imitate or execute human learning actions [12]. Nevertheless, enhancing the precision of present machine-learning algorithms, which have notable operational restrictions, is challenging.

The emergence of deep learning [13,14,15,16,17,18] has enabled the possibility of automatic and precise cell detection. The existing target recognition algorithms based on deep neural networks can be divided into single-stage [19] and two-stage [20].

The two-stage algorithms are represented by the RCNN [21], Faster R-CNN [22], SPP-Net [23], and FPN [24], which extract the region of interest (ROI) through the region proposal network (RPN), and then classify the region of interest. The two-stage algorithms implementing convolutional networks instead of fully connected layers reduce the number of unnecessary calculations inherent to the traditional sliding window approach. The two-stage algorithms need to be revised due to their inability to detect small targets and the high rate of mistakes.

The single-stage algorithms are represented by SSD [25] and YOLO [26,27,28], which realize end-to-end detection by considering localization and classification as a regression problem. The single-stage algorithms have fast detection speeds but low accuracy.

The detection results of blood cells under single-stage and two-stage algorithms, depicted in Figure 1, show that algorithms such as Faster R-CNN, SSD, and YOLO are prone to misdetection and omission.

Recently, some works [29,30,31] have begun to use the idea of an attention mechanism to improve the performance of computer vision tasks, such as image classification [32,33], object detection [34], and image segmentation [35]. Attention mechanisms in neural networks can divide into three types: channel attention mechanism [36], spatial attention mechanism [37], and mixed attention mechanism [38]. The channel attention mechanism focuses on more distinctive information by focusing on the channel relationship. In contrast, the spatial attention mechanism has advantages in extracting more detailed information, focusing on different spatial locations of feature images. The mixed attention mechanism combines the above two mechanisms and can simultaneously extract detailed and more distinctive information about feature species. Nevertheless, the attention mechanism usually requires more parameters. If the model is already complicated, adding the attention mechanism can result in overfitting and, thus, decrease the model’s effectiveness on the test set.

In summary, in this paper, we incorporate the Swin Transformer attention mechanism into YOLOv5s, which we call SDE-YOLO, and SDE-YOLO can solve the problems of insufficient detection accuracy and misdetection and omission in both single-stage and two-stage algorithms, as well as the issue of parameter computation in the attention mechanism. The innovation of this paper is as follows:Fusing the Swin Transformer [39] with the end of the backbone network can improve the feature extraction of small targets;Deleting the 32 × 32 down-sampling layer in PAN can reduce the number of network parameters while improving the detection rate;Replacing ordinary convolution with depth-separable convolution [40] in PANet can increase the detection of small targets while ensuring accuracy;Replacing the original CIOU loss function with the EIOU [41] loss function can better solve the problems of positive and negative sample imbalance while converging faster.

## 2. Related Work

### 2.1. YOLOv5s Model

YOLOv5 is divided into four models, s, m, l, and x, according to the depth and width [42], and, since blood cells are primarily small- and medium-sized targets, YOLOv5s is chosen as the basic framework for this paper. As shown in Figure 2, the network architecture of YOLOv5s is divided into four parts in total, as follows: Input, Backbone, Neck, and Prediction.

#### 2.1.1. Input

On the Input side, different sizes of input samples are scaled to a fixed size by adaptive image and then fed into the Backbone networks for training. The data enhancement of YOLOv5s adopts Mosaic data enhancement to solve the problems of insufficient samples and uneven distribution in the dataset. Mosaic data enhancement splices four images by random cropping, random brightness adjustment, and flipping, which reduces GPU use and enhances the network’s robustness.

#### 2.1.2. Backbone

The Backbone networks of YOLOv5s mainly consist of the CS, SPPF, and CBS modules, as shown in Figure 3a. The current Backbone networks in the field of target detection are generally ResNet [43], AmoebaNet [44], Xception [45], and other networks.

#### 2.1.3. Neck

The Neck parts of YOLOv5s adopt the structure of the feature pyramid network (FPN) [46] + the path aggregation network (PAN) [47], as shown in Figure 3b. The FPN structure passes the more vital semantic information from the more profound feature map to the shallower layer by up-sampling. The PAN structure transmits the location information from the superficial layer to the deep layer by down-sampling and performs multi-scale feature fusion.

#### 2.1.4. Prediction

The head of YOLOv5s uses three 1 × 1 convolutional layers to replace the fully connected layer (FC) for prediction and classification. It generates three different scales of feature maps, namely, 20 × 20, 40 × 40, and 80 × 80, and the outputs are 4 + 1 + C, which are the co-ordinates of the prediction frames, the target confidence, and the probability of the category values, respectively.

#### 2.1.5. Loss

The loss function mainly comprises the confidence loss $L_{\mathrm{obj}}$, classification loss $L_{\mathrm{cls}}$, and position loss $L_{\mathrm{obj}}$ of the target and predicted frames. The confidence loss and classification loss are obtained from the cross-entropy loss function, which is shown in Equation (1). The metric above is computed using the CIOU loss function, as in Equation (2). The total loss function is the sum of the three loss functions, as shown in Equation (3).
(1)$$L_{\mathrm{cls}}\left(p,y\right)=-\frac{1}{N}\overset{N}{\underset{i=1}{\sum }}\left[\mathrm{ylogp}+\left(1-y\right)\log\left(1-p\right)\right]$$
(2)$$\{\begin{array}{c} L_{\mathrm{obj}}=1-\mathrm{IOU}+\frac{\rho^{2}\left(b,b^{\mathrm{gt}}\right)}{c^{2}}+\alpha v\\ v=\frac{4}{\pi^{2}}\left(\arctan\frac{w^{\mathrm{gt}}}{h^{\mathrm{gt}}}-\arctan\frac{w}{h}\right)\\ \alpha =\frac{v}{1-\mathrm{IoU}\left(b,b^{\mathrm{gt}}\right)+v} \end{array}$$
(3)$$L_{\mathrm{total}}=L_{\mathrm{obj}}+L_{\mathrm{cls}}+L_{\mathrm{box}}$$

## 3. Proposed Method

### 3.1. Integration of the Swin Transformer Module

The Swin Transformer is based on the transformer with a global information modelling capability to construct hierarchical feature maps while drawing on the idea of the locality to limit the self-attention computation to the non-overlapping window area and allow moving windows for feature interaction, with linear computational complexity for the image size, which can be used as the backbone network of YOLOv5s to extract image features more effectively.

As shown in Figure 4, the Swin Transformer model builds four Swin Transformer block groups {C1, C2, C3, and C4} according to the number of Swin Transformer blocks [2,2,2,6], and the down-sampling multiplicity is [4,8,16,32], respectively. For the input image (H, W, 3), the patch partition layer divides it into 4 × 4 patches and converts the dimensionality of the image data into (H/4, W/4, 48). The linear embedding layer transforms the image data, the number of channels doubles, and the dimensions are (H/4, W/4, 96). In C1, the Swin Transformer block is based on the 7 × 7 window to calculate the multi-head self-attention and extract image features, and the output feature map has the exact dimensions of (H/4, W/4, 96). In the subsequent process, each patch merging layer first divides the input feature map into 2 × 2 patches, and then doubles the number of feature map channels to realize the down-sampling of the feature map. The dimension of the feature map transforms to (H/8, W/8, 192), (H/16, W/16, 384), and (H/32, W/32, 768), which is similar to the pooling operation in CNN. The Swin Transformer block groups of C2, C3, and C4 only extract the image data features without changing the feature map dimension.

The Swin Transformer block of the Swin Transformer consists of two modules: the window-based multi-head attention mechanism and mobile window-based multi-head attention mechanism. The information passes through the layer norm layer, window-based multi-head self-attention (W-MSA) unit, multilayer perceptron (MLP) layer, and jump connection of the left module to complete the multi-head self-attention calculation based on the window, and then passes through the Layer Norm layer, shifted window-based multi-head self-attention (SW-MSA) unit, MLP layer, and jump connection of the suitable module to complete the multi-head self-attention calculation based on the mobile window. The two modules have the same input and output information dimensions and are directly connected.

Window-based multi-head self-attention (W-MSA) models the dependency of a local window instead of the global one in multi-head self-attention (MSW), which helps to reduce the complexity of the model computation. W-MSA is similar to conv: it also has two parameters, the kernel size and stride, but W-MSA has the same kernel size and stride in the same stage, which leads to a problem—there is no difference between the receptive field of 1 W-MSA and n W-MSAs in a stage. A patch is applied to the W-MSA to get shifted window-based multi-head self-attention (SW-MSA) to alleviate this problem.

### 3.2. Improved the PANet Network Architecture

From the above, we know that the PANet network outputs predicted results by 8 × 8, 16 × 16, and 32 × 32 times, down-sampling for three small, medium, and large targets. Taking our blood cell BCCD dataset as an example, our size is 640 × 640 when we perform 8 × 8, 16 × 16, and 32 × 32 down-sampling. The output prediction layers are 80 × 80, 40 × 40, and 20 × 20, respectively. However, since the target sizes of our blood cell BCCD dataset are between 8 × 9 and 285 × 442, and the small- and medium-size targets take up 95% of the total cells, when the image is down-sampled by 32×, the blood cells smaller than 102 × 102 in the image will compress one pixel, which will lead to our inefficiency in small cell detection. In this paper, we have opted not to employ the 32 × 32 down-sampling correlation network layer, which will enable a faster cell detection process while reducing the number of parameters of the network model.

### 3.3. Improved Convolutional Structures in PAN

Deep separable convolution combines deep convolution and dot convolution (1 × 1 convolution). Deep convolution first performs a separate deep convolution for each input channel, and then mixes the output channels by dot convolution (1 × 1 convolution). The inputs are subject to filtering and combining operations in a standard convolution. Still, the deep separable convolution splits this operation by subjecting the information to a filtering operation in the first step, and then combining them in the second step. Composing a standard convolution reduces the network model’s computational cost and size. Figure 5 shows how a normal convolution operation decomposes a deep convolution and a dot convolution (1 × 1 convolution).

A feature map F of size $D_{F}\times D_{F}\times M$ is used as an input to a standard convolution. The output feature map G is $D_{G}\times D_{G}\times N$, where $D_{F}$ is the spatial width and height of the input feature map F, *M* is the number of input channels, $D_{G}$ is the spatial width and size of the output feature map G, and *N* is the number of output channels.

The computational cost $C_{conv}$ required for a standard convolutional layer is Equation (4), where the size of the convolution kernel is the square of $D_{K}$.
(4)$$C_{conv}=D_{K}\times D_{K}\times M\times N\times D_{F}\times D_{F}$$

The computational cost of deep convolution $C_{dconv}$ is Equation (5).
(5)$$C_{dconv}=D_{K}\times D_{K}\times M\times D_{F}\times D_{F}$$

The computational cost of deep convolution $C_{dconv}$ is Equation (6).
(6)$$C_{pconv}=M\times N\times D_{F}\times D_{F}$$

A linear combination of depth convolution and point-by-point convolution (1 × 1 convolution) is called depth-separable convolution. The computational cost of the depth-separable convolution is the sum of the computational cost of the depth convolution and the 1 × 1 point-by-point convolution as in Equation (7).
(7)$$C_{dsconv}=D_{K}\times D_{K}\times M\times D_{F}\times D_{F}+M\times N\times D_{F}\times D_{F}$$

Splitting the standard convolution into two steps, filtering and combining, can drastically reduce the amount of parameter computation, the computational cost of the depth-separable convolution, and the computational cost of the standard convolution, two specific parameter pairs, shown in Equation (8).
(8)$$\frac{C_{dsconv}}{C_{conv}}=\frac{D_{K}\times D_{K}\times M\times D_{F}\times D_{F}+M\times N\times D_{F}\times D_{F}}{D_{K}\times D_{K}\times M\times N\times D_{F}\times D_{F}}=\frac{1}{N}+\frac{1}{D_{K}^{2}}$$

When we use a deeply separable convolution with a convolution kernel of 3 × 3, the computational effort is one-eighth to one-ninth that of a standard convolution using the same-size convolution kernel, with essentially no difference in accuracy. By substituting the usual convolution in the PANet network with a deeply separable convolution, the detection of small targets improves, while the accuracy and real-time detection are maintained.

### 3.4. Improved IOU Loss Function

From the classification loss function in Equation (1), it can see that when the predicted probability value (*p* ≥ 0.5) is large, the loss value of the positive samples is smaller than that of the negative samples, which makes it difficult for the positive samples to influence the training parameters, thus affecting the detection accuracy. This paper adopts the EIOU loss function to solve the above problem, which is shown in Equation (9).
(9)$$L_{\mathrm{EIOU}}=L_{\mathrm{IOU}}+L_{\mathrm{dis}}+L_{\mathrm{asp}}=1-\mathrm{IOU}+\frac{\rho^{2}\left(b,b^{\mathrm{gt}}\right)}{c^{2}}+\frac{\rho^{2}\left(w,w^{\mathrm{gt}}\right)}{c_{w}^{2}}+\frac{\rho^{2}\left(h,h^{\mathrm{gt}}\right)}{c_{h}^{2}}$$

In the above equation: $L_{\mathrm{IOU}}$, $L_{\mathrm{dis}}$, and $L_{\mathrm{asp}}$ are the overlap loss, centroid loss, and width-height loss, respectively. b and $b^{\mathrm{gt}}$ are the prediction box and the real box, respectively. $\rho \left(b,b^{\mathrm{gt}}\right)$ is the Euclidean distance between the centroids of the two boxes, c is the diagonal distance of the smallest closure area that contains the two boxes, and $c_{w}$ and $c_{h}$ are the width and height of the smallest external box that covers the two boxes, respectively. w is the width of the prediction box, and $w^{\mathrm{gt}}$ is the width of the real box. h is the height of the prediction box, and $h^{\mathrm{gt}}$ is the height of the real box.

The overlap loss and center distance loss of the EIOU loss use the original method in CIOU. In contrast, the width–height loss directly minimizes the difference between the predicted and actual frames’ width and height, making convergence faster. Specifically, this loss function acts as a guide during the optimization process during training, causing the dimensions of the predicted frame to gradually approach the dimensions of the real frame. By incorporating the difference between the width and height of the predicted frame and the width and height of the real frame into the loss function, the model is more strongly constrained during the optimization process to predict the location and size of the target more accurately. This paper uses EIOU as the loss function of the improved model, which can effectively improve the accuracy.

## 4. Experiments

### 4.1. Dataset

This paper makes use of the relatively small BCCD dataset [48]. The BCCD dataset (Blood Cell Count and Detection Dataset) is a dataset open to the public, which is frequently employed in cell image analysis and medical image processing to count and detect blood cells. The primary purpose of this dataset is to facilitate researchers in constructing and assessing algorithms for the identification and enumeration of blood cells, thus enhancing the precision of medical diagnosis and treatment.

The BCCD dataset includes three varieties of blood cells: WBCs (white blood cells), RBCs (red blood cells), and Platelets. The data associated with each image, such as the location and number of cells, serve as a reference point for assessing the algorithm’s performance. The dataset consists of 768 images for training and 106 for testing, with the instances of three types of blood cells displayed in Figure 6.

### 4.2. Experimental Configuration

This paper’s experiments use Ubuntu 18.4.0 as the operating system. The experiments are conducted on the E-2136 CPU with a Quadro P5000 graphics card with 16 GB of video memory as its hardware configuration. The PyTorch 1.7.1 version is employed.

In Table 1, we can find the additional training parameters discussed in this paper.

### 4.3. Evaluation Indicators

In this research, the accuracy of the algorithm is assessed based on three factors [49]: experimental detection precision, recall [50], and mean average precision (mAP) [51], as presented in Equations (10)–(12).
(10)$$\mathrm{precision}=\frac{\mathrm{TP}}{\mathrm{TP}+\mathrm{FP}}$$
(11)$$\mathrm{recall}=\frac{\mathrm{TP}}{\mathrm{TP}+\mathrm{FN}}$$
(12)$$\mathrm{mAP}=\frac{1}{N}\int_{0}^{1}P_{i}\left(R\right)\mathrm{dR}$$

TP represents the number of true positives—instances when the model’s assessment is accurate, and the label is positive. FP represents the number of false positives—instances when the model’s assessment is wrong, and the label is positive. FN represents the number of false negative—instances when the model’s assessment is wrong, and the label is negative. TN represents the number of true negatives—instances when the model’s assessment is accurate, and the label is negative. The precision of a blood cell category measures the accuracy of the predicted category out of the total number of predictions. At the same time, the recall is the accuracy of the predicted category out of the total number of actual categories. The number of distinct blood cell types represents N in Equation (12), and the mean average precision of all the categories is mAP.

### 4.4. Results

The performance of SDE-YOLO for WBCs (white blood cells), RBCs (red blood cells), and Platelets is visualized in Figure 7 by plotting their respective P–R curves, and the mAP values of these blood cells obtained by computing the area beneath the curves.

The blood cell detection outcomes are presented in Figure 8b,c using the YOLO and SDE-YOLO algorithms, respectively. From the figure, we can see that the SDE-YOLO algorithm solves the problem of misdiagnosis of red blood cell as platelet in the first figure of Figure 8b, and the SDE-YOLO algorithm also solves the problem of missing red blood cells in the three figures of Figure 8b. In summary, the SDE-YOLO algorithm can solve the misdetection and omission problems in the original single-stage and two-stage algorithms.

The TP confusion matrix we generated is shown in Figure 9, from which we can see that, in addition to the background, red blood cells, white blood cells, and platelets form a diagonal matrix, which is ideal.

Table 2 compares SDE-YOLO with some single-stage and two-stage detection algorithms.

We can see from Table 2 that we achieved higher mAP and FPS than all other detection algorithms except YOLOv5s. SDE-YOLO has a higher mAP than Faster-R-CNN [22], YOLOv3 [26], YOLOv4 [27], TE-YOLOF-B3 [52], and ISE-YOLO [53], by 19.5, 12.1, 11.3, 4.1, and 10.3 percentage points, respectively. SDE-YOLO has a higher FPS than Faster-R-CNN, YOLOv3, YOLOv4, TE-YOLOF-B3, and ISE-YOLO, by 34.2, 8.9, 7.3, 0.4, and 8.9, respectively.

SDE-YOLO has a lower FPS than YOLOv5s [42] due to the Swin Transformer’s integration and deep separable convolution incorporation, resulting in a 13.3 FPS difference. But SDE-YOLO has the mAP that is 6.4 percentage points greater than YOLOv5s.

The experimental results indicate that the SDE-YOLO algorithm is the most effective of the models examined, as it improves the mAP without compromising the detection speed.

The graph in Table 3 shows the performance of SDE-YOLO compared to YOLOv7 and YOLOv8.

SDE-YOLO is compared with YOLOv7. The F1_curve is the same as YOLOv7; both are 0.92. SDE-YOLO is 1.3% less than YOLOv7 in terms of precision, but SDE-YOLO is 0.1% more than YOLOv7 in terms of mAP, and 31.3 more in terms of FPS than YOLOv7.

Compared with YOLOv8, SDE-YOLO is 0.7% less than YOLOv8 in mAP, 11.4% less than YOLOv8 in FPS, and 1% less than YOLOv8 in F1_curve. However, SDE-YOLO is 1.3% higher than YOLOv8 in terms of accuracy.

SDE-YOLO has its advantages in performance over both YOLOv7 and YOLOv8, and all three models meet the accuracy and speed requirements.

## 5. Comparison with Other Datasets

We also assess SDE-YOLO on additional datasets to corroborate its generalization. The first dataset is the LIDC-IDRI set [54], a medically related collection of lung nodules, and we obtained 3144 images for the training set and 349 images for the test set through annotation. The TT100K dataset [55] comprises traffic signs, most of which are small- or medium-sized, and annotates 8816 images from the training set and 1712 images from the test set. The graphical representation of the two datasets’ performance metrics is in Figure 7.

The experimental results show that SDE-YOLO achieves a mAP of 72% on the LIDC-IDRI dataset, which is higher than the 70% of YOLOv5s. As illustrated in Figure 10a, SDE-YOLO exhibits a significant performance improvement compared to YOLOv5s when applied to the LIDC-IDRI dataset.

The mAP of SDE-YOLO on the TT100K dataset is comparable to YOLOv5s, which are both 96%, and Figure 10b shows that the final curve of SDE-YOLO is gradually approaching YOLOv5s.

In conclusion, SDE-YOLO has demonstrated its effectiveness across various datasets.

## 6. Ablation Studies

We run ablation experiments to assess the impact of each component on SDE-YOLO, with YOLOv5s as the foundation. A represents the combination of Swin Transformer, B represents the excision of the 32 × 32 network layer, C signifies the implementation of depth-separable convolution, and D indicates the use of the EIOU loss function to raise the effectiveness. The findings of the experiment are displayed in Table 4.

Table 4 indicates that individual enhancement points contribute to the model’s performance index, and that amalgamating multiple enhancement points is advantageous for the model. Compared to YOLOv5s, mAP improved by 4.1% to 7.5%, F1_curve was enhanced by 0.1% to 0.3%, and accuracy improved by 0.2% to 6.4%.

## 7. Conclusions and Discussion

This paper presents a new algorithm, SDE-YOLO, which improves upon YOLOv5s. To begin with, we present an overview of the YOLOv5s architecture. Next, this paper outlines four new ideas: the integration of the Swin Transformer, the deletion of the 32 × 32 target layer in PANet, the substitution of standard convolution with depth-separable convolution in PANet, and the use of the EIOU loss function in place of the CIOU loss function. This paper evaluates the single-stage and two-stage algorithms of the past, as well as YOLOv7 and YOLOv8, respectively, by conducting comparison experiments. The SDE-YOLO algorithm has demonstrated its effectiveness on other datasets, and finally, we test the impact of its components through ablation experiments. The run shows that SDE-YOLO is accurate and can be implemented swiftly.

However, during the implementation of blood cell testing, one may encounter the following challenges: 1. Quality and quantity of datasets: Blood cell image datasets may be affected by image quality, resolution, cueing conditions, and other factors. Obtaining large-scale and diverse blood cell datasets is challenging and crucial for model training and generalization; 2. Category imbalance: There may be an imbalance in the number of different types of cells in the blood cell images, resulting in a degradation of the performance of a few categories in the model. Strategies are needed to balance different categories of samples to improve the detector’s ability to recognize various blood cells; 3. Geometric variations: The sizes and shapes of blood cells may have more significant variations in the image. The model must adapt to cells with various geometries and can detect targets with different geometries; 4. Overlap and duplication: Blood cells may overlap and duplicate, etc., which makes target detection more urgent.

Future research will explore the fusion of multiple data sources to improve detection accuracy. Utilizing multimodal information can allow for a more comprehensive detection of blood cells and enhance the ability of the model to capture targets at different scales.

## Figures and Tables

**Figure 1 biomimetics-08-00404-f001:**
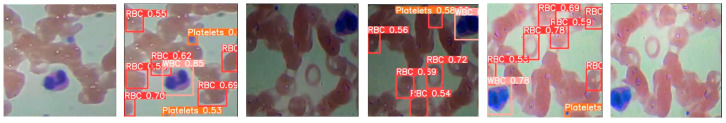
Comparison of blood cells under single-stage and two-stage algorithms.

**Figure 2 biomimetics-08-00404-f002:**
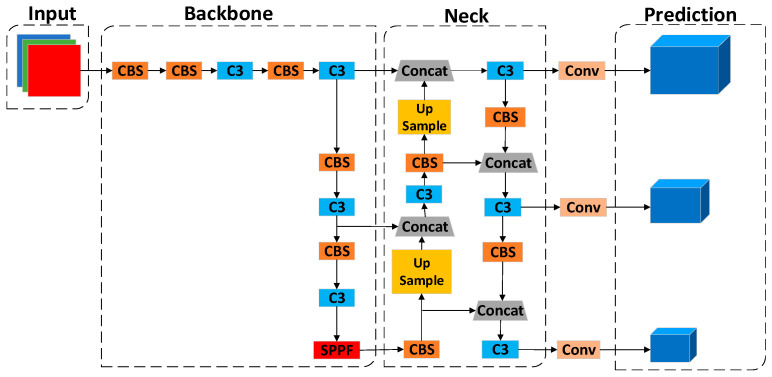
YOLOv5s network architecture.

**Figure 3 biomimetics-08-00404-f003:**
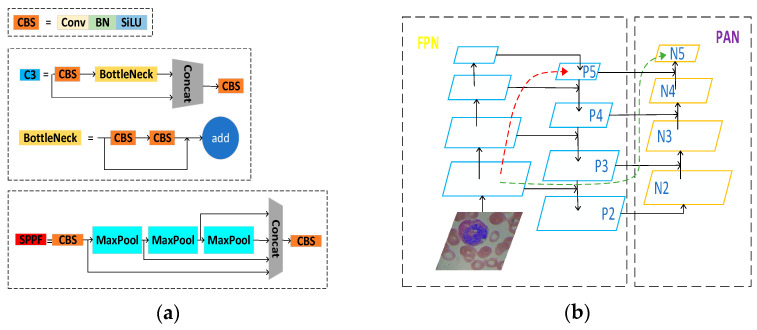
(**a**) Represents the main modules of YOLOv5s backbone network; (**b**) Representing the structure of the neck network.

**Figure 4 biomimetics-08-00404-f004:**
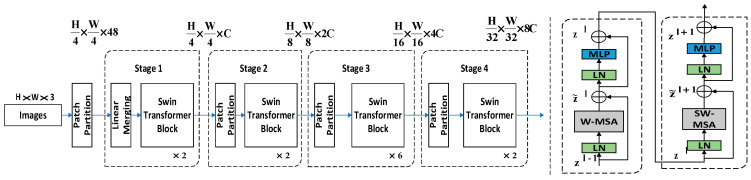
Swin Transformer network architecture.

**Figure 5 biomimetics-08-00404-f005:**
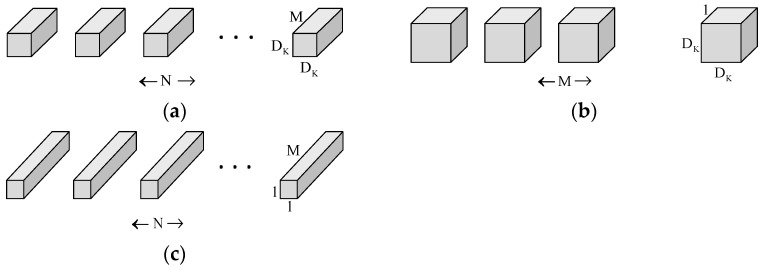
(**a**) Representing the filters for standard convolution; (**b**) Representing the filter for deep convolution; (**c**) Representing the filter for pointwise convolution.

**Figure 6 biomimetics-08-00404-f006:**
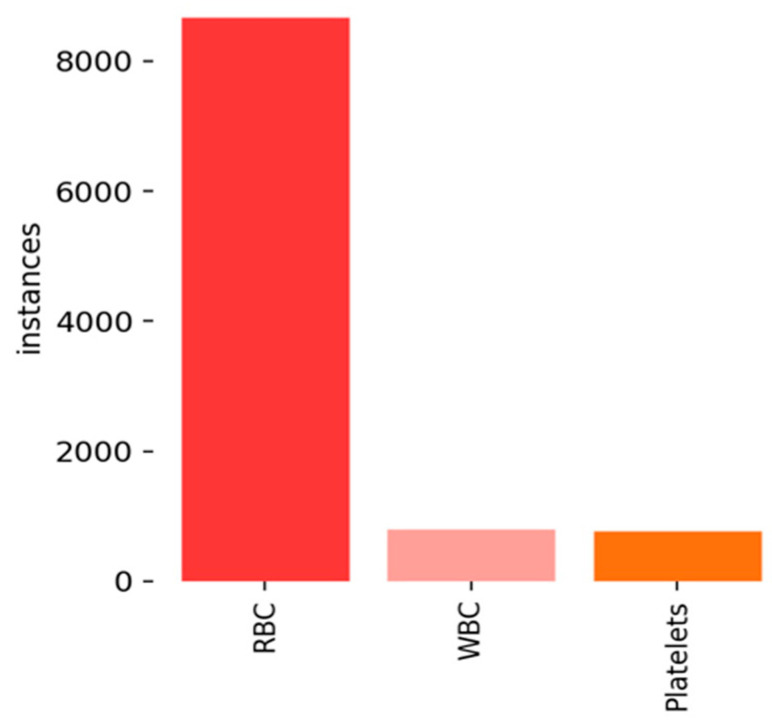
Instances of three types of blood cells.

**Figure 7 biomimetics-08-00404-f007:**
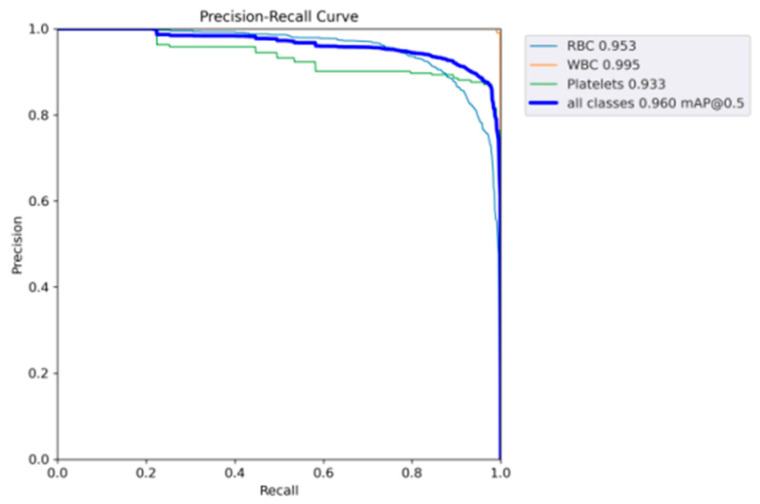
Representing the P–R curves of SDE-YOLO.

**Figure 8 biomimetics-08-00404-f008:**
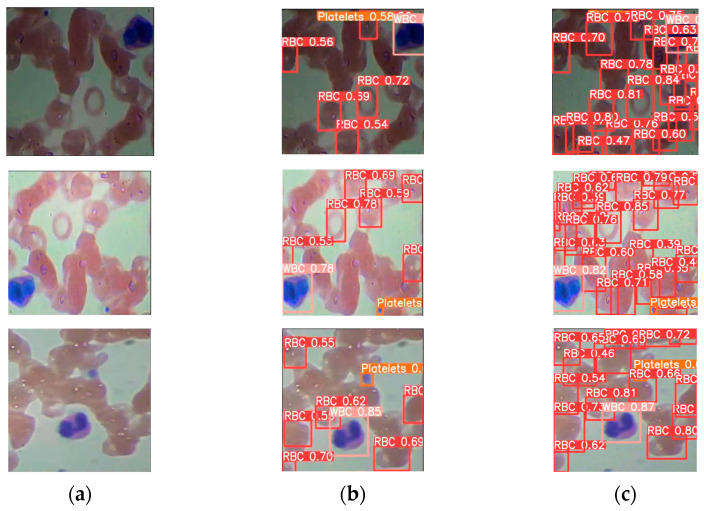
Representing the P–R curves of SDE-YOLO; (**a**) Original image of blood cells; (**b**) Blood cell images under other YOLO algorithms; (**c**) Blood cell images under SDE-YOLO detection algorithm.

**Figure 9 biomimetics-08-00404-f009:**
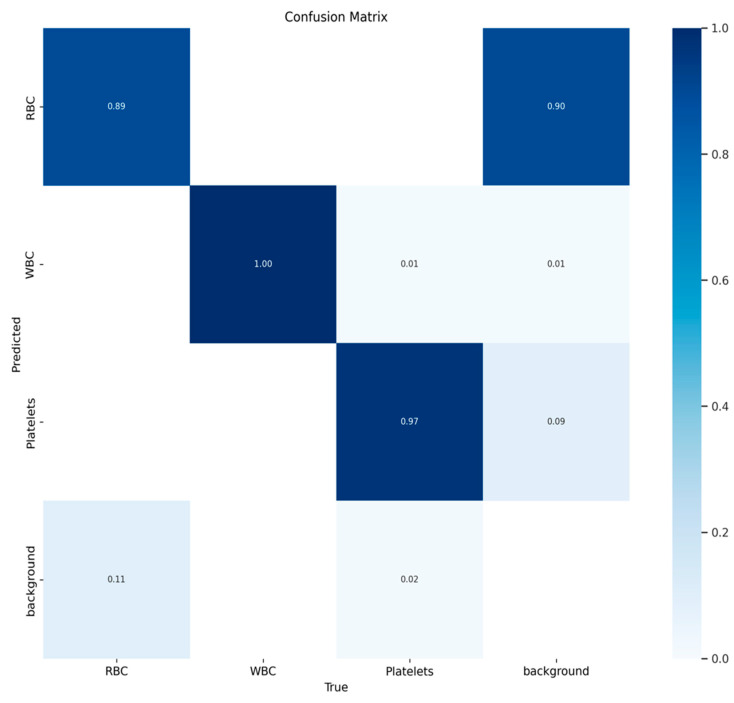
Confusion matrix results for blood cells.

**Figure 10 biomimetics-08-00404-f010:**
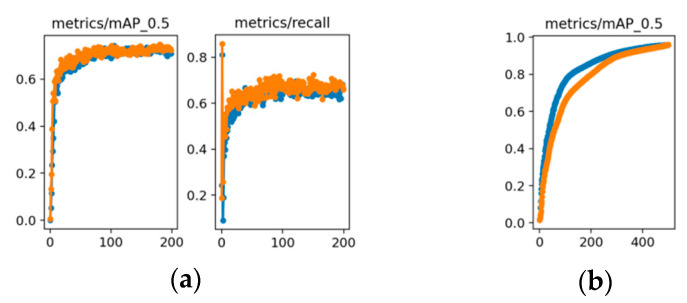
(**a**) Comparison of the YOLOv5s algorithm and the SDE-YOLO algorithm on the LIDC-IDRI dataset; (**b**) Comparison of the YOLOv5s algorithm and the SDE-YOLO algorithm on the TT100K dataset. The blue line represents the original YOLOv5s algorithm and the orange line represents the SDE-YOLO algorithm.

**Table 1 biomimetics-08-00404-t001:** Training parameter settings for the BCCD dataset.

Parameter Name	Parameter Values	Parameter Name	Parameter Values
Batch size	16	Epoch	133
Learning rate	0.01	Momentum	0.937
Box loss	0.05	Cls loss	0.5
Anchors	(10, 13), (16, 30), 33, 23), (30, 61), (62, 45), (59,119)	Obj loss	1.0

**Table 2 biomimetics-08-00404-t002:** Comparative analysis experiments.

Method	Input Size	WBCs	RBCs	Platelets	mAP	FPS
Fast-RCNN [22]	1000 × 600	0.803	0.722	0.770	0.765	9.2
YOLOv3 [26]	608 × 608	0.914	0.829	0.774	0.839	34.5
YOLOv4 [27]	640 × 640	0.930	0.798	0.813	0.847	36.1
TE-YOLOF-B3 [52]	416 × 416	0.987	0.873	0.898	0.919	43
ISE-YOLO [53]	416 × 416	0.965	0.927	0.896	0.857	34.5
YOLOv5s [42]	640 × 640	0.977	0.838	0.873	0.896	56.7
SDE-YOLO	640 × 640	0.995	0.953	0.933	0.960	43.4

**Table 3 biomimetics-08-00404-t003:** Experiments with the latest technology.

Method	Input Size	WBCs	RBCs	Platelets	mAP	FPS	F1_curve	Precision
YOLOv7	640 × 640	0.995	0.954	0.928	0.959	9.1	0.92	0.864
YOLOv8	640 × 640	0.995	0.960	0.945	0.967	54.8	0.93	0.807
SDE-YOLO	640 × 640	0.995	0.953	0.933	0.960	43.4	0.92	0.851

**Table 4 biomimetics-08-00404-t004:** Ablation studies.

Method	Input Size	WBCs	RBCs	Platelets	mAP	F1_curve	Precision
YOLOv5s [42]	640 × 640	0.977	0.838	0.873	0.896	0.90	0.858
YOLOv5s + A	640 × 640	0.995	0.949	0.959	0.968	0.93	0.922
YOLOv5s + B	640 × 640	0.991	0.938	0.953	0.961	0.92	0.892
YOLOv5s + C	640 × 640	0.995	0.959	0.945	0.966	0.92	0.865
YOLOv5s + D	640 × 640	0.995	0.950	0.967	0.971	0.93	0.888
YOLOv5s + A + B	640 × 640	0.957	0.899	0.955	0.937	0.91	0.889
YOLOv5s + A + C	640 × 640	0.995	0.957	0.958	0.970	0.93	0.898
YOLOv5s + A + D	640 × 640	0.995	0.950	0.953	0.966	0.93	0.920
YOLOv5s + B + C	640 × 640	0.995	0.953	0.954	0.967	0.92	0.970
YOLOv5s + B + D	640 × 640	0.995	0.945	0.954	0.965	0.92	0.861
YOLOv5s + C + D	640 × 640	0.995	0.937	0.945	0.959	0.91	0.888
YOLOv5s + A + B + C	640 × 640	0.995	0.948	0.953	0.965	0.92	0.862
YOLOv5s + A + C + D	640 × 640	0.995	0.948	0.955	0.966	0.93	0.909
YOLOv5s + B + C + D	640 × 640	0.995	0.939	0.958	0.964	0.92	0.904
SDE-YOLO	640 × 640	0.995	0.953	0.933	0.960	0.92	0.902

## Data Availability

The data used to support the findings of this study are included in this article.

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
