# Peer review of "SDE-YOLO: A Novel Method for Blood Cell Detection"

_biomimetics, 2023, doi:10.3390/biomimetics8050404_

Round 1

Reviewer 1 Report

In this manuscript, the authors propose a modification to YOLOv5's architecture with the goal of enhancing the detection accuracy of platelets, white and red blood cells. The authors performed their work in a systematic manner by including benchmark results, ablation studies, and by showing that their results can be generalized to other datasets.

However, the model developed by the authors (SDE-YOLO) does not outperform state-of-the-art algorithms (i.e., YOLOv8) in terms of neither detection speed nor accuracy. Nevertheless, their approach may still be of interest to readers as long as the authors clearly explain in a detailed manner how their approach can be used in other applications (with specific examples) and/or how the rationale utilized to develop SDE-YOLO could be useful to further enhance the performance of state-of-the-art algorithms, such as YOLOv8.

There is a high frequency of typos and unclear/confusing sentences throughout the text. The authors should carefully proofread their manuscript and extensively edit it. A few examples:

Line 114 (typos): "The is calculated from the CIOU loss function"

Line 282-283 (unclear sentences and word repetition): "Although some performances are not as good as theirs, some performances are still better than theirs, and all of them can satisfy the requirements of accuracy and real time."

Line 293: "We experimentally get that the mAP of SDE-YOLO is 72% on the LIDC-IDRI dataset"

Author Response

Thank you for taking the time to review our work and provide insightful feedback. We truly appreciate your expertise and thoughtful comments, which will undoubtedly help enhance the quality of our research. We now attaching our detailed response. 

Reviewer 2 Report

In this paper, authors proposed an improved target detection algorithm SDE-YOLO based on YOLOv5s framework, which firstly integrates the Swin Transformer at the end of the backbone to extract the features in a better way. Then removes the 32 x 32 network layer in PANet to reduce the number of parameters of the network while better recognizing small targets. In addition, it replaces the normal convolution with depth-separable convolution in the PANet to better identify small targets while ensuring real-time performance. Finally, replacing the CIOU loss function with the EIOU loss function can improve the positive and negative sample imbalance problem and accelerate the convergence speed. The authors did good work and interested for the readers. The following review comments are recommended, and the authors are invited to explain and modify.

1 Abbreviations should be given full name in abstract.

2 What is practical application of proposed SDE-YOLO model?

3 The introduction section needs to be improved. An introduction is an important road map for the rest of the paper that should be consist of an opening hook to catch the researcher's attention, relevant background study, and a concrete statement that presents main argument but your introduction lacks these fundamentals, especially relevant background studies. This related work is just listed out without comparing the relationship between this paper's model and them; only the method flow is introduced at the end; and the principle of the method is not explained. To make soundness of your study must include these latest related works.

I (2021). Deep learning-based predictive identification of neural stem cell differentiation. Nature Communications, 12(1), 2614. doi: 10.1038/s41467-021-22758-0

II (2023). Stimulated Raman Scattering Microscopy Enables Gleason Scoring of Prostate Core Needle Biopsy by a Convolutional Neural Network. Cancer Research, 83(4), 641-651. doi: 10.1158/0008-5472.CAN-22-2146

III (2022). Progressive Distributed and Parallel Similarity Retrieval of Large CT Image Sequences in Mobile Telemedicine Networks. Wireless communications and mobile computing, 2022. doi: 10.1155/2022/6458350

IV (2022). Improved Feature Point Pair Purification Algorithm Based on SIFT During Endoscope Image Stitching. Frontiers in Neurorobotics. doi: 10.3389/fnbot.2022.840594

4 “Width-height loss directly minimizes the difference between the predicted and actual frames' width and height, making convergence faster”, need details.

5 How to optimize hyperparameters during model training shown in Table 1?

6 When writing phrases like “experimental detection precision (precision), recall (recall), and mean average precision (mAP)”, it must cite related works in order to sustain the statement 10.1155/2022/2665283; 10.1155/2023/2345835.

7 TP denotes the number of correct positive cases; confusion matrix needs to be shown.

8 Dataset need detailed information.

9 Mention the limitations and future works of the developed system elaborately.

10 Authors should mention the implementation challenges.

Minor editing of English language required.

Author Response

Thank you for taking the time to review my work and for your insightful comments and suggestions. Your feedback is invaluable, and I greatly appreciate your expertise and dedication to the review process. We now attaching our detailed response

Round 2

Reviewer 2 Report

We appreciated the authors' efforts in manuscript revision, and the following minor concerns need to be discussed and revised carefully before the paper's acceptance.

1 Figure 4. Swin Transformer Network Architecture needs a detailed description.

2 Please show high-quality Figure 8. Confusion matrix results for blood cells.

3 When writing phrases like "experimental detection precision (precision), recall (recall), and mean average precision (mAP)", it must cite related works in order to sustain the statement (10.1155/2022/2665283; 10.1155/2023/2345835); these mentioned works used the same evaluation metrics, but the authors did not include them to sustain the statement.

4 Figure 7 needs a detailed description of the phenomenon that happened there and also needs improved image quality.

5 Moreover, it should be noted that the clinical appliance has to be decided by medicals since the existing differences between the real image and the one generated by the proposed model could be substantial in the medical field.

Minor editing of English language required.

Author Response

1 Figure 4. Swin Transformer Network Architecture needs a detailed description.

Respond:

Thank you for your valuable comments, and we acknowledge that our Swin Transformer section is not adequately presented and may have confused the reader.

In our latest revision of the manuscript, we have added some clarifications, including the full spelling of S-MSA and SW-MSA in English and an introduction to S-MSA and SW-MSA to make it easier for readers to read.

Please let us know if this revision meets your expectations. We appreciate your hard work to make the content complete.

2 Please show high-quality Figure 8. Confusion matrix results for blood cells

Respond:

Thank you for drawing our attention to this issue. We apologise for the lack of clarity of the results graph in the paper. In the latest revision of the results, we have enlarged the results to make them more understandable to the reader.

Please let us know if this revision meets your expectations, we appreciate your hard work.

3 When writing phrases like "experimental detection precision (precision), recall (recall), and mean average precision (mAP)", it must cite related works in order to sustain the statement (10.1155/2022/2665283; 10.1155/2023/2345835); these mentioned works used the same evaluation metrics, but the authors did not include them to sustain the statement.

Respond:

Thank you for bringing this to our attention. We apologize for any oversight in not providing proper citations for the mentioned metrics. We have revised the text accordingly to ensure that the sources for the experimental detection precision (precision), recall (recall), and mean average precision (mAP) are properly referenced. The relevant references have been cited as the two you recommended.

Please let us know if this revision meets your expectations, we appreciate your hard work.

4 Figure 7 needs a detailed description of the phenomenon that happened there and also needs improved image quality.

Respond:

Thank you for bringing this issue to our attention. We apologise for the lack of elaboration of the test results and the lack of clarity of the results.

In this latest revision, we have added a description of the advantages of SDE-YOLO over other one-stage and two-stage algorithms, and enlarged the results graphs to make them more understandable to the reader.

Please let us know if this revision meets your expectations, we appreciate your hard work.

5 Moreover, it should be noted that the clinical appliance has to be decided by medicals since the existing differences between the real image and the one generated by the proposed model could be substantial in the medical field.

Respond:

Thank you for reminding us of the problems we may encounter when applying the model. We acknowledge that clinical application must be decided by medical staff and we will certainly extract doctors' opinions carefully when applying the model. In addition, we will further improve the dataset and the quality of the dataset to make it as close as possible to the real data. And we will try to improve the generalisation ability of the model.